# Collision Detection and Avoidance for Underwater Vehicles Using Omnidirectional Vision [note 1]

**DOI:** 10.3390/s22145354

**Published:** 2022-07-18

**Authors:** Eduardo Ochoa, Nuno Gracias, Klemen Istenič, Josep Bosch, Patryk Cieślak, Rafael García

**Affiliations:** Computer Vision and Robotics Research Institute (VICOROB), University of Girona, 17003 Girona, Spain; nuno.gracias@gmail.com (N.G.); klemen.istenic@gmail.com (K.I.); jep250@gmail.com (J.B.); patryk.cieslak@udg.edu (P.C.); rafael.garcia@udg.edu (R.G.)

**Keywords:** visual SLAM, omnidirectional multi-camera systems, collision risk assessment, risk map, ROVs, AUVs

## Abstract

Exploration of marine habitats is one of the key pillars of underwater science, which often involves collecting images at close range. As acquiring imagery close to the seabed involves multiple hazards, the safety of underwater vehicles, such as remotely operated vehicles (ROVs) and autonomous underwater vehicles (AUVs), is often compromised. Common applications for obstacle avoidance in underwater environments are often conducted with acoustic sensors, which cannot be used reliably at very short distances, thus requiring a high level of attention from the operator to avoid damaging the robot. Therefore, developing capabilities such as advanced assisted mapping, spatial awareness and safety, and user immersion in confined environments is an important research area for human-operated underwater robotics. In this paper, we present a novel approach that provides an ROV with capabilities for navigation in complex environments. By leveraging the ability of omnidirectional multi-camera systems to provide a comprehensive view of the environment, we create a 360° real-time point cloud of nearby objects or structures within a visual SLAM framework. We also develop a strategy to assess the risk of obstacles in the vicinity. We show that the system can use the risk information to generate warnings that the robot can use to perform evasive maneuvers when approaching dangerous obstacles in real-world scenarios. This system is a first step towards a comprehensive pilot assistance system that will enable inexperienced pilots to operate vehicles in complex and cluttered environments.

## 1. Introduction

The study of marine habitats is of utmost importance for the underwater scientific community. For many decades, marine habitats have been explored in search of hydrothermal vents or for seabed characterization in the deployment of offshore oil and gas infrastructures. However, since many of these applications involve environments that are inaccessible and/or dangerous for direct human intervention, the development of underwater robots such as remotely operated vehicles (ROV) and autonomous underwater vehicles (AUV) plays an important role.

The applications and use of underwater robots have been recently growing and they are becoming an integral part of current marine operations, with high-accuracy navigation, planning, and mapping in these systems playing a vital role. Therefore, it is not a surprise that recent research is directed towards achieving better hardware and software capabilities. Several underwater applications require a combination of sensors, such as surface Global Positioning Systems (GPS), inertial navigation system (INS), Doppler velocity logger (DVL), multibeam echosounder (MBES), or laser scanners, which can obtain results that vary in precision and performance. However, certain limitations that these types of sensors have in water and the high costs of fiber optics INS, as well as the growing capabilities of visual sensors, have led the community to start developing new algorithms based on simultaneous localization and mapping (SLAM) systems along with cameras as the main exteroceptive sensor when the robot operates close to the seabed.

Furthermore, in the last few years, omnidirectional cameras have received increasing interest from the computer vision community in tasks such as augmented reality, motion estimation, and SLAM. Advantages of omnidirectional 360° multi-camera system (MCS), such as their wide field of view (FOV), their high-resolution and high-speed image acquisition, and their lower costs have opened the door to new technological applications. The wide FOV is very convenient for visual simultaneous localization and mapping (vSLAM) and mapping tasks, especially in the case of ROV, as they allow the pilot to operate the robot directly through images captured by the omnidirectional cameras. This new capability will extend the pilot’s spatial awareness and reduce the common orientation problems during missions, particularly in confined or cluttered environments where pilots need to be highly aware of the environment to ensure the safety of the robot.

Omnidirectional camera systems enable a comprehensive 3D reconstruction of the surroundings, which can support spatial and proximity awareness in all directions. As such, the envisioned system described in this paper is intended to generate warnings when an ROV pilot reaches a position of high collision risk and to override human inputs sent to the vehicle control. This can be seen as a first step towards a comprehensive driver assistance system that would enable inexperienced pilots such as scientists to operate vehicles in complex environments. This concept is shown in Figure 1.

## 2. Related Work

The following sections provide a short overview of closely related literature in the areas of visual SLAM, omnidirectional cameras systems, and underwater collision avoidance.

### 2.1. Visual SLAM

The detection and avoidance of obstacles in a given environment is linked, inherently, to the process of estimating the position of a robot with respect to each object around it. At any time, a fully autonomous system needs a local representation of the surroundings to analyze potential risks. Such a system can be divided into two processes: the localization and mapping of the robot and its environment, and a collision detection and avoidance process. The first process has been researched over a few decades and is generally known as SLAM. Many algorithms have been proposed to achieve this task, which vary in methods, and the sensors used to acquire data. When cameras are employed as the only exteroceptive sensors, it is referred to as vSLAM. Over the past few years, there has been a trend in using visible radiation as the only external perception modality, and the vSLAM problem has been extensively studied [1,2,3,4,5]. This trend is partly due to the ability of optical cameras to obtain range and appearance information about the environment, and the low cost of the equipment. One of the pioneering solutions was proposed by Davison [6]. The proposed filter-based vSLAM algorithm, named MonoSLAM, was demonstrated using a single monocular camera to extract features from the environment and estimate the position by using an extended Kalman filter (EKF). Since then, a variety of solutions using different sensor configurations have been proposed including monocular [6,7,8,9,10], stereo [11,12,13], omnidirectional [14,15,16], and combinations of cameras with other sensors. Additionally, the depth of solutions can be categorized by the method’s approach into feature-based [6,7], direct approaches [13,14,17], and keyframe approaches [8,11,12,18]. In the past decade, vSLAM approaches such as ORB-SLAM [8,11,12] have been incrementally improved to work with several sensor configurations (e.g., monocular, stereo, RGB-D cameras, and visual–inertial system configurations) and to be able to robustly estimate the position of a robot.

### 2.2. Omnidirectional and Multi-Camera Systems

More recently, the SLAM community has benefited greatly from the development and increased accessibility of large FOV cameras and multi-camera systems. This type of system configuration can cover a wider field of view, leading to better pose-tracking robustness due to the observation of more reliable features. Unfortunately, the number of full SLAM contributions using multiple cameras has been very limited, as opposed to monocular and stereo configurations. Kaess and Dellaert [19] made use of an eight-camera rig in a general, non-stereo setting, to reconstruct the full 360° view of a mobile robot in an indoor office environment. On the other hand, Zou and Tan [20] studied the vSLAM problem in dynamic environments with multiple cameras. Their Collaborative SLAM (CoSLAM) implementation runs in real time (38 ms per frame) using inter-camera tracking and mapping. It also uses methods to distinguish static background points from dynamically moving foreground objects. Multi-camera PTAM (MCPTAM) [21,22] is a different approach where the authors changed the perspective camera model to the generic polynomial model. The system can be used with cameras with minimal or non-overlapping FOV. Urban et al. [18] presented a multi-camera vSLAM system, MultiCol-SLAM, which made use of the generic camera model and was developed as an extension of the state-of-the-art algorithm ORB-SLAM [8]. The implementation was released as an open-source code [23] and introduced the concept of multi-keyframes (MKF) and multi-camera loop closure, among other performance improvements.

### 2.3. Collision Avoidance

As applications in complex and unstructured environments are growing, autonomous systems are being required to carry out missions safely and efficiently. To accomplish this, the system needs to be able to correctly sense, detect, and make decisions in case of an impending collision. Collision avoidance has been addressed and researched over the last decades, but it is still a trending topic in robotics applications, e.g., unmanned aerial vehicles (UAV), AUV, and advanced driver assistance system (ADAS). The first stage of the collision avoidance problem, that is, sensing the surrounding environment, has usually been carried out using different types of proprioceptive and exteroceptive sensors mounted in autonomous vehicles (AV). The use or combination of GPS, LIDAR, cameras, ultrasonic sensors, and other sensors can be found in the literature. The second stage, which involves detecting collisions with static or dynamic objects, relates to the approach used to model the system and its environment. A distinction between collision detection systems should be made depending on their goal. On one hand, research on approaches to detect whether a collision has happened or not, and how to model it, can be found in [24,25,26,27,28,29]. On the other hand, a collision-avoidance system focuses its interest on the prediction of when and where a collision might happen.

Predicting when a collision might happen closely relates to path planning schemes, and research has been carried out in this field [30,31,32,33,34]. However, unlike path planning approaches, collision avoidance usually refers to the ability of the vehicle to recognize dangers and act simultaneously. Generally, Collision avoidance systems (CAS) are navigation systems with only a local understanding of the surroundings. This makes them reactive systems based on instantaneous information rather than global motion planner systems. One of the first real-time approaches to achieve such reactive behavior was the potential field method (PFM) proposed by Khatib [35]. This method generates virtual repulsive forces from obstacles to repel the robot away from them. However, this method assumes a known and predefined world model of the obstacles. Later techniques, such as the vector field histogram (VFH) [36] and the dynamic window approach (DWA) [37], formulate the collision avoidance problem in unknown environments, therefore allowing for a more reactive behavior. A pioneering implementation of this type of system in real time was presented by Borenstein et al. [38], based on the paradigm of virtual force field (VFF), where certainty grids were used for obstacle representation, and navigation was achieved through the use of potential fields. More recent research, such as [39,40], employ sensors, such as laser scanners and cameras, to model the robot’s surroundings and navigate through unknown environments.

Vision-based obstacle avoidance has been extensively used due to the high data rates that cameras provide and their relatively low cost. Flacco et al. [41] used an optical depth sensor, such as the Microsoft Kinect, to acquire depth data and compute distances between the robot and the obstacles. The distances are used to generate repulsive forces to control the behavior of the robot. A vision-based collision avoidance system for UAVs was proposed by [42], using a dynamic potential field to repel the vehicle away from hazards. Perez et al. [43] made use of a stereoscopic system to compute a disparity map from images provided by the stereo camera and located zones free of obstacles to guide a multi-copter onto collision-free trajectories.

Apart from the ability to sense, locate, and react to obstacles, intelligent vehicles should be able to detect dangerous situations and react accordingly. While the research studies, described above, propose techniques and models to plan a collision-free trajectory, they do not assess how dangerous an obstacle may be and whether or not it is necessary to act upon its presence. This component of CAS is commonly known as risk assessment. Risk represents the likelihood and severity of the damage that a vehicle may suffer on the path ahead of it, and it can be assessed by different methods, the most common ones being trajectory prediction, distance estimation, probabilistic methods, and using risk surrogates [44,45].

Risk surrogates, such as the time to collision (TTC) and time to react (TTR), are popular indicators of how critical a potential collision may be. Ammoun et al. [46] presented an approach for collision risk estimation between vehicles by assessing the risk with respect to the TTC and three other risk indicators. In their approach, the vehicles used positional information provided by GPS receivers and communication devices to predict the trajectories of other surrounding vehicles and identify possible collisions. An image-based approach to assess the collision risk with TTC was proposed by Pundlik et al. [47]. Their system used the local scale change of the images and the motion of the obstacles to predict the object’s trajectory relative to the camera. In [48], a real-time low-cost system was proposed for ADAS using monocular video. The system estimates the future automotive risk of a driving situation by using object tracking based on a particle filter over an intermediate time horizon. Probabilistic-based approaches for risk assessment can be found in [49,50,51]. Moreover, with the rise of artificial intelligence (AI) and deep neural networks (DNN), approaches using deep predictive models for collision risk assessment are being currently researched [52,53].

### 2.4. Collision Avoidance in Underwater Robotics

Having a fully autonomous system is a primary goal in all areas of robotics, including underwater applications. To date, research carried out with underwater vehicles has focused on path-planning algorithms [54,55,56,57,58,59] and collision avoidance systems [60,61,62,63,64] using different types of sensors, such as multi-beam echo-sounders, laser scanners, and profiling sonars. Hernández et al. [55] proposed a system that merged the information of multiple sensors to create a system able to map, plan, and direct missions autonomously. The map created by the system was used to recompute collision-free paths in real time, according to the obstacles perceived. In later work, they extended their approach to create photo-realistic textured 3D models [56]. In [62], a collision avoidance method, based on the collision risk assessment and an improved velocity-obstacle method, was presented to allow unmanned underwater vehicles to perform accurate evasive decisions in dynamic environments. Palomeras et al. [58] proposed a probabilistic methodology to explore complex environments, with no information known a priori. The method uses a scanning sensor and a repetitive algorithm based on selecting generating several candidate viewpoints from the vehicle’s current position, pruning them down according to a series of criteria, and evaluating the utility of each candidate viewpoint. After determining the viewpoint with the greatest utility, an obstacle-free path to the viewpoint is generated and navigation to this point is carried out, where a new scan is taken.

Although effective systems have been proposed to avoid obstacles and reach a goal, and some research has been carried out using cameras for underwater applications [65,66,67], few ongoing studies of vSLAM and multi-camera systems applied to ROV or AUV robots have been developed. In [68], a reactive navigation system was used to guide the exploration of an AUV along obstacle-free trajectories. The acquired images are used to detect and segment regions of water apart from those with obstacles, which then are used to compute collision-free routes. Wirth et al. [69] presented the integration of two stereo visual odometry algorithms into an AUV to estimate the linear movement and rotation and be able to navigate close to the seabed. Some other algorithms, such as [70,71], make use of image information and deep neural networks to provide AUVs with avoidance schemes. Manderson et al. [71] proposed a real-time navigation system that exploits visual feedback to make close-range navigation possible. The convolutional neural network used in the algorithm can predict unscaled and relative path changes, which are transformed into absolute paths, thus allowing the vehicle to navigate close to structures and avoid dangers. Our paper departs from the existing literature by exploring the capabilities of MCS. We describe a method which uses an omnidirectional camera along with an SLAM system to create dense map representations of the AUV surroundings. The localization of the robot is computed with an open-source vSLAM system as the backend (MultiCol-SLAM). This localization information, along with an assessment of the risk, is passed to a collision avoidance control system, to execute a reactive evasive maneuver in case of danger.

## 3. Contributions

Nowadays, the most advanced AUVs are guided, commonly, by systems that use sensors such as echo-sounders, GPS, LIDAR, laser scanners, or a fusion of many of these, to perform path planning and collision avoidance when navigating in unknown environments. However, there is little ongoing research on collision avoidance using optical sensing in underwater environments. Those that exist are mainly based on a monocular or stereo system [69,70,71] which do not take advantage of the capabilities that a MCS can provide to an AUV. The main contributions of the present work are the following:We developed a new camera-based omnidirectional collision avoidance system that exploits the capabilities of multi-camera setups to perform a 360° real-time 3D reconstruction of the region surrounding an ROV/AUV, which allows the robot to assess the risk presented by local objects and act accordingly. A system such as this can carry out survey missions in a more autonomous way while ensuring the safety of the robot.A framework based on an open-source omnidirectional vSLAM package that can reconstruct a map of the surveyed environment, and assess in real-time how dangerous the surrounding objects are. To our knowledge, this is the first time an omnidirectional system has been applied to underwater environments.A set of warning signals that can assist operators during exploratory missions. These outputs provide an intuitive way of knowing where obstacles are and they allow the operator to perform evasive maneuvers. This also allows less-trained pilots to operate the vehicles safely, thus reducing operational costs.

This work represents an extended version of preliminary work presented in [72], where the main idea was presented as a proof-of-concept, without validation on a real robotic platform.

## 4. Approach

### 4.1. Framework

Our proposed system performs a 360° 3D reconstruction of the surroundings to reliably assess the risk of collision with any nearby objects. In case of potential danger, it provides warning signals, which can be easily interpreted by a pilot, or by the control system of an autonomous vehicle, to perform evasive maneuvers.

The proposed framework uses omnidirectional images from an MCS as input to a vSLAM system. The SLAM system estimates the motion of the robot in order to pass this information to the collision detection thread. This thread uses the estimated pose and MKFs (i.e., a set of selected key images that stores the camera images and the corresponding pose) to compute a denser point cloud of the local map, which is assessed to estimate the collision risk of each 3D point in the map. Our system is capable of providing two outputs calculated from the estimated risk values:An estimate of the repulsion force which would cause the robot to move away from a potential collision;An omnidirectional risk map.

While the repulsion force is well suited as a control input for both AUVs or ROVs, the omnidirectional 2D equirectangular risk map is intended to provide ROV pilots with visual tools to easily understand and perceive the zones where potential collisions could happen. Figure 2 shows the described system.

#### 4.1.1. Multi-Camera Tracking System

The SLAM system used in our solution was based on MultiCol-SLAM [18], which consists of three main threads: tracking, local mapping, and loop closing, which operate in parallel at all times. Since collision detection is only relevant in the present, it requires a local map that accurately describes the structure around the robot’s current position; therefore, the noisy and sparse map created during the SLAM process is not suitable. To overcome this, vSLAM is used as precise visual odometry that provides a reliable estimate of relative position, which is used later by the collision detection thread to assess the risk of surrounding obstacles.

#### 4.1.2. Tracking and Mapping

The tracking thread processes each new frame coming from the MCS. The tracking is performed by extracting features from every image of the MCS and using them to identify matches across MKFs. The current pose is estimated by using the relative orientation between the last two frames. If the tracking does not provide a sufficient number of matches, a re-localization step must be performed. The tracking thread also decides if it is time to add a new MKF to the map. This decision is implemented if one of the following conditions are met:More than 0.5 s (half of the acquisition period of time) have passed since the last MKF insertion.A certain number of poses must be successfully tracked since the last re-localization. This number is set to the current frame rate, as in [18].At least 50 points are tracked in the current pose.If the visual change is big enough (i.e., less than 90% of the current map points are assigned to the reference MKF).

If the tracking thread decides to insert a new MKF, the local mapping thread extends the map by creating new points and deleting all redundant map points and MKFs. The deletion of map points is performed if they are visible in fewer than three MKFs, and an MKF is deleted if any pair of MKFs shares more than 90% of the same map points. Furthermore, the mapping thread also maintains the consistency of the global map by performing a global bundle adjustment optimization step.

### 4.2. Collision Avoidance

The collision detection thread is the central contribution of this work and runs concurrently with the SLAM system. The tasks being performed during execution time are depicted in Figure 2. The process begins as soon as the SLAM system’s tracking thread can reliably estimate the relative pose between two sequential MKFs. The collision thread first creates a local dense map describing the robot’s environment using the dense map reconstruction process described in Section 4.2.1. The risk of collision is subsequently assessed by taking into account the trajectory and speed of the AUV.

Since the processes of extracting the local 3D information and risk assessment are based on pairs of MKFs, it is required that there are no drastic changes in either speed or direction of the vehicle that could significantly alter the usability of the estimated risk. Given that the SLAM system ensures a dynamic selection of MKFs, the time difference between the two images is always small enough and proportional to the movement of the vehicle so this can be considered a reasonable assumption.

#### 4.2.1. Dense Map Reconstruction

The main goal of the dense map reconstruction process is to use the previously estimated relative pose between two sequential MKFs to obtain a dense representation of the vehicle’s surroundings. The dense map can therefore be computed by using the poses of both MKFs to calculate their relative transformation matrix before performing triangulation and reconstructing the scene. This process consists of the following steps:Image preprocessing: Due to the lack of texture in underwater scenarios, the use of filtering techniques helps to extract more features from the scene.Feature detection and matching: Feature points can be extracted using any feature extractor and subsequent matching is performed by exploiting epipolar constraints obtained from the relative transformation matrix.Scale estimation: A scale correction procedure is applied to the initial 3D reconstruction before the 3D information is used for risk assessment. This process is necessary as MultiCol-SLAM can occasionally produce pose estimations with inaccurate scale estimates due to the inherent scale ambiguity of SfM-based techniques. The scale is corrected by taking measurements from the AUV odometry. By comparing the real displacement performed by the robot and the displacement given by the SLAM system from time t−1 to *t*, a scale correction factor can be calculated and multiplied by the transformation matrices. This ensures that the 3D information is obtained in metric units.Triangulation: The 3D point cloud is obtained by computing the projection matrices of both frames and by performing a triangulation in which the outliers are rejected based on the re-projection error.

### 4.3. Risk Estimation

The risk of each point in the dense map is defined as the probability that the point will collide with the vehicle, given the speed and trajectory of the vehicle. In other words, it assesses how potentially harmful a point can be to the robot. This assessment is performed by using the TTC risk surrogate, which indicates the time required by any point to reach the robot’s current position.

Furthermore, to correctly assess the risk of any point with respect to the current position, we have to consider that the estimated pose (which comes from SLAM) is referenced to the camera location. Thus, the assessed risk of a point represents the probability of collision with the camera. Therefore, to determine the real value of the risk, we defined a set of surrounding points located at critical points of the AUV, and the risk is calculated by evaluating each of these critical locations. These points are assigned taking into account the geometry of the AUV, and the number of total points and their positions are set at the points where a collision is expected to occur first.

As previously mentioned, the risk estimation is performed by acquiring a time measure that describes how far a certain obstacle is from the robot. The TTC value is obtained by considering the equations of a point in 3D to a line [73] (Figure 3), where the line represents the current heading trajectory of our AUV. Let our heading trajectory line be represented by two robot’s point positions in space $r_{1}^{t-1}$ and $r_{1}^{t}$ lying on it, so this vector is given by
(1)$$v=\left[\begin{array}{c} x_{1}+\left(x_{2}-x_{1}\right)t\\ y_{1}+\left(y_{2}-y_{1}\right)t\\ z_{1}+\left(z_{2}-z_{1}\right)t \end{array}\right]$$
where $\left(x_{1},y_{1},z_{1}\right)$ and $\left(x_{2},y_{2},z_{2}\right)$ correspond to the 3D coordinates of the points $r_{1}^{t-1}$ and $r_{1}^{t}$, respectively.

Thus, time *t* corresponds to the time required for the robot to reach the projection of a 3D point, $x_{n}$, onto this line (Figure 3). Furthermore, the squared distance between a point in the line with parameter *t* and $x_{n}$ can be found as
(2)$$d^{2}={\left[\left(x_{1}-x_{0}\right)+\left(x_{2}-x_{1}\right)t\right]}^{2}+\;{\left[\left(y_{1}-y_{0}\right)+\left(y_{2}-y_{1}\right)t\right]}^{2}+\;{\left[\left(z_{1}-z_{0}\right)+\left(z_{2}-z_{1}\right)t\right]}^{2}$$

Subsequently, the time *t* that minimizes the distance between the point $x_{n}$ and the line can be found by derivation of Equation (Equation 2). Therefore, *t* is defined as
(3)$$t=-\frac{\left(r_{1}^{t-1}-x_{n}\right)\cdot \left(r_{1}^{t}-r_{1}^{t-1}\right)}{{∥r_{1}^{t}-r_{1}^{t-1}∥}^{2}}$$

Finally, the minimum distance, *d*, of a point $x_{n}$ to the line can be found by substituting Equation (Equation 3) in Equation (Equation 2).
(4)$$d=\frac{∥\left(r_{1}^{t}-r_{1}^{t-1}\right)\times \left(r_{1}^{t-1}-x_{n}\right)∥}{{∥r_{1}^{t}-r_{1}^{t-1}∥}^{2}}$$

The value of the distance is also taken into account in the risk computation, as points further away from our heading trajectory must be assigned a lower risk than those closer to it.

As mentioned, time *t* (Equation (Equation 3)) and the minimum distance *d* (Equation (Equation 4)) are required to assess the risk of any $x_{n}$ point in the local map. However, as these equations are defined for a specific point $r_{i}$ of the AUV structure, we extended this computation for all points defined to form a security cage around the robot. Figure 4 shows how these measurements are taken into account for a representative number of points $r_{i}$ on the body of the robot. Thus, when the AUV makes a displacement from t−1 to *t*, the measures *d* and *t* are redefined as
(5)$${}^{r_{i}}t_{x_{n}}=-\frac{\left(r_{i}^{t-1}-x_{n}\right)\cdot \left(r_{i}^{t}-r_{i}^{t-1}\right)}{{∥r_{i}^{t}-r_{i}^{t-1}∥}^{2}}$$
(6)$${}^{r_{i}}d_{x_{n}}=\frac{∥\left(r_{i}^{t}-r_{i}^{t-1}\right)\times \left(r_{i}^{t-1}-x_{n}\right)∥}{{∥r_{i}^{t}-r_{i}^{t-1}∥}^{2}}$$

In order to take into account the velocity of the vehicle, all these measurements are additionally normalized by the time elapsed between the two MKFs. For each point in the scene, ${}^{r_{i}}{\mathrm{TTC}}_{x_{n}}$ and ${}^{r_{i}}d_{x_{n}^{o}}$ are therefore considered to be
(7)$${}^{r_{i}}{\mathrm{TTC}}_{x_{n}}={(}^{r_{i}}t_{x_{n}}-1)\cdot \Delta t$$
(8)$${}^{r_{i}}d_{x_{n-norm}}=\frac{{}^{r_{i}}d_{x_{n}}}{∥{}^{r_{i}}d_{x_{n}}∥}\cdot \Delta t$$

Since the total time is calculated starting from the previous position $r_{i}^{t-1}$, a simple subtraction of a unit time step (t=1) is required to set the system into the current frame. Equations (Equation 7) and (Equation 8) therefore express the normalized values of TTC and the minimum distance in seconds.

Finally, from our previous work [72], the risk is calculated using a series of Gaussian functions with zero mean and peak values centered on the movement trajectory. The standard deviations, σ, of these functions are defined as linear functions of TTC times and a free parameter *k*. The higher the value of *k*, the higher the risk assessment becomes, increasing caution for objects further away as the risk cone opens further around the current direction of motion. As a result, objects that are not directly on the trajectory line (i.e., side objects) are perceived with a higher risk. Figure 5 shows the effects that changing the parameters *k* or TTC have on the risk function. The risk for each 3D point $x_{n}$ with respect to a given point $r_{i}$ on the robot’s chassis is defined as
(9)$${}^{r_{i}}\rho_{\left(x_{n}\right)}=\frac{k}{\sigma }e^{-\frac{{{(}^{r_{i}}d_{x_{n}})}^{2}}{2\sigma^{2}}}\;\;\;\;\;\;;\;\;\;where\;\;\;\;\sigma =k\cdot {(}^{r_{i}}{\mathrm{TTC}}_{x_{n}})$$

Finally, the risk value is chosen as the one with the highest value among those calculated for all selected points on the robot chassis:(10)$$\rho_{\left(x_{n}\right)}=max{(}^{r_{i}}\rho_{\left(x_{n}\right)})$$

### 4.4. Avoidance Scheme

#### 4.4.1. Resultant Repulsive Force

Once the risk of each 3D point is computed, the final task of the system is to use this information to send warning signals and compute the repulsion force that is required to move the robot away from a potential collision. These signals are used to perform avoidance maneuvers away from zones where the danger of collisions is high, as well as to alert ROV pilots. Given that the number and density of detected 3D points in the environment can vary significantly, a sphere of 3D points $s_{j}$ is created surrounding the robot position, as illustrated in Figure 6a. This ensures a consistent estimation of a repulsive force independent from the number and density of 3D points. Furthermore, for all the detected points $x_{n}$ in the local map, a repulsion force vector is computed coming from each point in the opposite direction of the viewing ray. This direction is calculated with respect to the corresponding point where the risk has the maximum value $r_{i,max(\rho_{x_{n}})}$:(11)$$\vec{f_{x_{n}}}=\frac{r_{i,max(\rho_{x_{n}})}-x_{n}}{∥r_{i,max(\rho_{x_{n}})}-x_{n}∥}$$

The computed repulsive vectors $\vec{f_{x_{n}}}$ and the corresponding vector to each sphere point $s_{j}$ are used to compute the final repulsive force required to act on the system.

As presented in Algorithm 1, the computation of the resultant repulsive force is performed in three steps. First, each repulsive force $\vec{f_{x_{n}}}$ of all 3D points is compared to all the vectors, coming from the same point on the robot, towards the sphere points $\vec{s_{j}}$. The comparison is made by computing the cosine of the angle between both vectors as
(12)$$cos{\left(\alpha \right)}_{x_{n},s_{j}}=\frac{\vec{f_{x_{n}}}\cdot {\vec{s}}_{j}}{∥\vec{f_{x_{n}}}∥∥{\vec{s}}_{j}∥}$$
where *n* represents each of the triangulated 3D points and *j* a sphere point.

After computing the value of Equation (Equation 12) for all 3D points, we associate each $\vec{f_{x_{n}}}$ with the vector $\vec{s_{j}}$ where cos(α) has the maximum value. This association allows us to relate 3D points to a certain area of the sphere. Then, its corresponding risk values are added to this area in the sphere. The final risk value of each sphere point $s_{j}$ comes from the sum of the risks of all contributing 3D points, given by
(13)$$\rho_{s_{j}}=\sum_{}^{}risk\;of\;each\;x_{n}\;associated$$

Taking into consideration that the contribution of 3D points to all areas on the sphere is not uniformly distributed, the risk values of the sphere points should be normalized. A threshold value setting the minimum number of contributing 3D points is also used to perform risk normalization. This avoids the case where outliers may be the only contribution to the risk of a specific sphere point. The normalization is defined as
(14)$$\bar{\rho_{s_{j}}}=\frac{\rho_{s_{j}}}{M}$$
where $\bar{\rho_{s_{j}}}$ corresponds to the mean value of the risk for a point $s_{j}$ in the sphere. The parameter *M* corresponds to the total number of points $x_{n}$ that contribute to a point in the sphere $s_{j}$ or to a threshold value, which is set to a value of five points in the case that not many points $x_{n}$ contribute to this area.

Finally, repulsive forces for each sphere point $s_{j}$ are computed by multiplying its normalized vector by their corresponding risk values and, subsequently, combined into a single repulsive force by summing all individual forces. This is performed by using Equations (Equation 15) and (Equation 16). The described algorithm above is also briefly depicted in Figure 7.
(15)$$\vec{F_{s_{j}}}=∥\vec{s_{j}}∥\cdot \bar{\rho_{s_{j}}}$$
(16)$$\vec{F}=\sum_{}^{}\vec{F_{s_{j}}}$$

**Algorithm 1** Resultant repulsive force computation
**Input:**
$\vec{f_{x_{n}}}$: repulsive force vectors$\rho_{\left(x_{n}\right)}$: assessed risk values$s_{j}$: 3D points defined on a surrounding sphere around the robot
**Output:**
$\bar{F}$: Resultant repulsive force
  1:Load the discretized sphere of points $s_{j}$  2:**for** each point $s_{j}$ in the sphere **do**  3:   $M_{j}\leftarrow 0$ {counter for the number of contributing points}  4:   $\rho_{s_{j}}\leftarrow 0$  5:
**end for**
  6:**for** each 3D point $x_{n}$ **do**  7:   **for** each point $s_{j}$ **do**  8:     Compute the $cos{\left(\alpha \right)}_{x_{n},s_{j}}$ between the vector of each point of the point cloud $x_{n}$ with the vector of each point in the surrounding sphere $s_{j}$  9:   **end for**10:   Find the maximum value of $cos{\left(\alpha \right)}_{x_{n},s_{j}}$, which represents the vector on the sphere that is closer to the vector $x_{n}$11:   $\rho_{s_{j},max\left(cos\alpha \right)}\leftarrow$ Add the corresponding $risk_{x_{n}}$ to the risk of the found point $s_{j}$ of the sphere12:   $M_{j,max(cos\alpha )}\leftarrow M_{j}+1$13:
**end for**
14:**for** each point $s_{j}$ in the sphere **do**15:   **if** $M_{j}$
**not** 0 **then**16:     Normalize the risk value of the point $\rho_{s_{j}}$ by the total number of contributing points $M_{j}$17:   **else**18:     Normalize the risk value of the point $\rho_{s_{j}}$ by a set threshold $M_{minimum}$19:   **end if**20:   $\vec{F_{s_{j}}}\leftarrow ∥\vec{s_{j}}∥\cdot \bar{\rho_{s_{j}}}$21:
**end for**
22:

$\vec{F}\leftarrow \sum_{}^{}\vec{F_{s_{j}}}$




#### 4.4.2. Equirectangular Risk Map Visualization

The system also generates a 2D representation of the risk in the scene. This is performed by re-projecting 3D points onto a spherical 360° panorama, represented by the equirectangular form [74]. The equirectangular representation of the entire viewing sphere is a 2D image in which the horizontal axis represents azimuth angles from 0° to 360° and the vertical axis represents altitude angles from −90° to +90° [65].

To create a complete risk map the following steps were carried out:Projecting 3D points to the image: Each 3D point with coordinates Q=(X,Y,Z) is first converted to spherical representation:
(17)$$R=\sqrt{X^{2}+Y^{2}+Z^{2}}$$
(18)θ=atan2(Y,X),0≤θ≤2π
(19)$$\phi =acos(\frac{Z}{R}),0\leq \phi \leq \pi$$
where (R,θ,ϕ) denote the spherical coordinates of point *Q*, i.e., the radial distance, polar angle, and azimuthal angle. The final image coordinates (u,v) of point *Q* are computed through Equation (Equation 20).
(20)$$u=\frac{\theta +\pi }{2\pi }*W\;\mathrm{and}\;\;v=\frac{\phi }{\pi }*H$$
where *W* and *H* are the desired width and height of the equirectangular image.Smoothing: Since the direct projection of points from the dense 3D map leads to an unclear representation with spurious calculated risks in some areas, an additional step of interpolation of the missing results is required. Smoothing is carried out on a unit sphere to avoid the problems of performing it directly on the equirectangular images. Points with known risk values are projected onto this sphere. Then, the risk for each pixel of the equirectangular image is calculated by projecting them onto the same sphere. Finally, the average risk of the *N* nearest points with known risk, weighted by their distance from the projected pixel, is calculated. To limit the spread of risk to areas without known information, the interpolation is limited to pixels within a certain distance from the nearest point with known risk.

## 5. Results

The system capabilities were assessed in two phases of experiments. The first phase, or offline testing, used a set of images captured by an ROV during an exploratory mission. In the second phase, the system was assessed using a simulated environment on the Stonefish simulator [75] and also by integrating the system into a real AUV, the GIRONA1000, and performing tests in a controlled environment. This latter phase is the online testing. All the results are described and discussed below.

### 5.1. Offline Test

#### 5.1.1. Dataset

The offline phase tested the main capabilities of the system with a dataset consisting of images of a survey over a WWII shipwreck sunk close to the shore of Palamós, Spain, named Boreas. The data were obtained with the SPARUS II AUV of the University of Girona. The robot was equipped with an omnidirectional camera composed of five GoPro Hero 4 Black edition cameras [74]. This MCS is capable of recording video at 30 fps with a resolution of 27 Mpixels at up to 200 m depth. Additionally, to acquire extremely close-range imagery, the robot was guided by divers to safely steer it through narrow passages. This configuration is shown in Figure 8.

During offline tests, we simplified the problem by modeling the robot and the MCS as a single point in space, located at the origin of the MCS system. Thus, the chassis of Sparus II was not considered in the risk calculation. Using the simplified system, we can test our framework to extend it to complex configurations later.

#### 5.1.2. Effects of Parameter *k* on Risk Computation

The effects of varying the free parameter *k*, at constant velocity, on the risk calculation were evaluated and can be visualized in Figure 9. Two types of representations are visualized in the figure. The first representation (left) shows the distribution of locally tracked points around the robot with their associated risk value represented by a color scale value. Points at larger distances (e.g., at the sides or further away) are given a lower weight when risk is computed. The second representation (right) is the equirectangular map output of the system, with the same colored risk representation.

The results show that increasing the value of *k* gives more weight to objects farther away from the robot. Figure 9a,b depict this behavior. When *k* is changed from 1.2 to a value of 2, objects that are on the sides of the heading direction (distances greater than 10 normalized units) are given corresponding risks between 0.4 and 0.5. However, the same points were much less dangerous in the first scenario (Figure 9a), with risk values between 0.2 and 0.35. We can also observe that, in the equirectangular map representation of Figure 9b, zones such as the railing of the boat are now considered a high-risk area (red area). This demonstrates that *k* allows us to have control of the cautioning window and can be adjusted to consider side objects.

#### 5.1.3. Effects of Vehicle Speed

The speed of the robot is also a variable that we analyzed in our tests. Larger speed values of the ROV mean a lower reaction time to obstacles, which translates into higher risk values for objects at a given position. Figure 10 shows the risk values of locally tracked points at the same position in time, but with different velocities. To test this, the value of *k* is kept constant (at a value of 1.2) and the timestamp of successive MKFs were modified. Figure 10a shows the effect of modeling the system with a velocity equal to four times the value of Figure 9a, and Figure 10b represents a velocity of half the original value, i.e., values of 1/4Δt and 2Δt, respectively.

The effect of changing the speed of the system is better illustrated in the distribution diagrams in Figure 10 (left). When the velocity is increased to four times its original value, the distribution of the tracked points shifts to the left and they cluster around distance values from 0 to 10 units, which means that they are now perceived as highly dangerous objects. Therefore, the equirectangular map representation now has red risk zones in the heading direction. If, on the other hand, the speed is halved (Figure 10b), the same distribution of points in the diagram is more widely spread. Since the system is now moving slowly, the tracked local points are no longer perceived as dangerous because it has more time to react; therefore, the risk values are lower.

#### 5.1.4. Resultant Repulsive Force Behavior

The resultant repulsive force was computed at each time step, and its behavior on the system was analyzed. Figure 11 illustrates one trajectory performed by Sparus II in the Boreas shipwreck, and multiple time steps are shown throughout its entire movement. At any position, the behavior of the computed resultant force (orange vectors on the image) always points outwards from zones where the biggest danger is perceived. Since the risk is dependent on the heading direction, the force vector is also dependent on the direction of movement and the risk associated with local obstacles. Therefore, objects perceived outside of the cautioning window and at the back of the robot have only a small contribution to the reactive behavior of the force.

#### 5.1.5. Ground Truth Comparison

The performance of the system, i.e., how well it predicts and reacts to collisions, was evaluated by comparing the calculated equirectangular risk maps against precomputed equirectangular depth images, which are used as ground truth. These maps were previously calculated using structure from motion (SFM) and multi-view stereo algorithms.

The difference between the system output and the precomputed SFM risk map is used as the performance measure. The resulting image is analyzed to identify the discrepancies between our predictions and the ground truth. Values of this difference where our algorithm risk prediction is high but where the ground truth is zero lead to a false positive (FP) result. This is represented in Figure 12 as red zones in the difference image. On the other hand, the blue zones on the image represent false negative (FN) areas where our risk estimation was calculated as low but which in reality should have higher values.

Figure 13 shows the variability of the performance of our system. The total values of FP and FN were computed for each frame, and summed to obtain their resultant value in the risk map. Then, the percentage was computed by dividing it by the whole image area. It can be seen that the mean values of FP and FN are approximately 2% and 8%, respectively, which indicates a low percentage of the image area. Some outliers can be found where these values increase, but the range of the errors for both measures do not exceed the value of 10% for most of the cases (at least for 75% of the data).

### 5.2. Real-Time Simulation Testing

#### 5.2.1. Realistic Simulated Experiment in *Stonefish*

The first real-time validation of the developments was performed using our open-source *Stonefish* C++ simulation library [75] combined with the ROS interface package, called *stonefish_ros*. This software is specifically designed for the simulation of marine robots. It delivers full support for rigid body dynamics of kinematic trees, geometry-based hydrodynamics, buoyancy, and collision detection. It also simulates all types of underwater sensors and actuators to seamlessly replace the real system with a simulated robot. Moreover, its modern rendering pipeline delivers realistic underwater images with light absorption and scattering models. The full software architecture of the Girona1000 AUV was used, following the hardware-in-the-loop (HIL) simulation paradigm. The simulation scenario included a textured patch of underwater terrain and the robot itself. The robot was equipped with an omnidirectional camera, composed of six pinhole cameras. Several tests were performed using a representative number of points on the Girona1000 AUV chassis to calculate the risk values for the surrounding 3D points.

#### 5.2.2. System Performance in Simulation

Real-time performance was tested with the Stonefish simulator to recreate an exploratory mission in an underwater environment. To achieve this, an ROS package was created. The ROS package manages the incoming set of images from the simulated omnidirectional camera system and processes them to obtain the corresponding system output. Finally, the repulsion force outputs are used by the control system to perform evasive maneuvers when necessary.

In the initial offline tests, we modeled our robot as a point in space, and the risk calculations were performed in relation to this point. However, to meet real-world conditions and to ensure the safety of the robot, we later modeled the robot body as a seven-point structure that took into account the extreme points where the robot could collide with an object while performing any movement, thus creating a safety cage for our robot. Figure 14 shows the different views of Girona1000 AUV with the designed safety cage, shown in green. The risk of all 3D points in the local map was calculated against all seven cage points, and the maximum risk among them is considered to calculate the corresponding repulsion force of each 3D point of the local map.

When the entire Girona1000 structure is taken into account for calculating the risk of all 3D points triangulated at a given time, several danger zones can occur simultaneously while the robot is exploring the terrain. Figure 15 illustrates this situation. The figure shows that when the robot approached one of the walls of the environment, two hazard zones were obtained from the risk calculation process. These zones correspond to the lower part of the Girona1000 chassis and the structure of the camera system, as seen in Figure 15b, which is a magnified view of Figure 15a.

In addition, to ensure robot safety, the force calculated from the risk of each 3D point is also sent to a control system that moves the robot out of danger zones. Whenever the robot approaches a wall or an obstacle, the magnitude of this force starts to increase and when it reaches a threshold value, it triggers an evasive maneuver coming from the control system. Figure 16 shows the behavior of the system when Girona1000 is near an obstacle. In Figure 16b, we can observe the configuration of the system at one time step before the magnitude of the force becomes greater than the danger threshold. The force points away from the zone, which is mainly yellow with some red points. However, since the direction of the course has not changed, one time step later, the red zone increases and the density of red points make the repulsion force magnitude rise above the set threshold. At this point, a signal is sent to the control system, which takes control of the movement of the Girona1000. This is represented in Figure 16c by the fact that the safety cage turns red, which means that even if the operator tries to tele-operate the Girona1000, the control system will not accept these inputs until system safety is guaranteed. Finally, Figure 16d shows the point in time when the control system stops maneuvering and the control of the robot is returned to the pilot, indicated by the safety cage returning to its green color.

### 5.3. Real-Time AUV Deployment

#### Experimental Setup

The framework was tested for real-time validation on an AUV. The robot chosen was the GIRONA 1000 AUV, which has the advantage that it can be reconfigured for different tasks by changing its payload and the configuration of the thrusters. In addition, the robot was reconfigured to carry an underwater omnidirectional multi-camera system (OMS) based on a Point Grey’s Ladybug 3 camera [65]. The system consists of six individual cameras, five of which (the side cameras) have their optical centers on the same plane and the last camera points in the direction normal to the plane. To use the camera in underwater environments, the camera was encapsulated inside a custom waterproof housing, made ofpoly-methyl methacrylate (PMMA), which makes it submersible to a water depth of 60 m (Figure 17). As a result, the factory calibration was no longer valid and a new calibration procedure was required. The specific calibration procedure is described in [65].

The tests and deployment of the robot were performed in a controlled environment, our 16 × 8 m testing pool with a progressive depth of 5 m. On the other hand, to validate the capabilities of the robot, the pool was provided with textured images to simulate an underwater environment. Objects were distributed so that the robot can navigate through them without colliding. The testing pool characteristics and the layout for the testing can be visualized in Figure 18.

### 5.4. Camera Housing Image Distortion Correction

One of the most important aspects to take into account is the presence of strong refraction of the optical rays due to the interfaces between media, i.e., a ray of light coming from the water changes its direction twice before reaching the sensor as it must pass through water–PMMA and PMMA–air transitions. To accurately model the distortion due to this effect, it becomes essential to model and simulate the intersection of each light ray with the different media. The direction of the refracted ray can be computed through Snell’s law [76]. From previous work [65], we computed a set of 3D rays and their corresponding set of 3D points on the outer surface of the custom housing. Having this data and knowing both the intrinsic and extrinsic parameters of all the cameras, it is possible to use the ray-tracing approach to project any 3D point underwater onto any of the cameras.

As a first step, to correct the distortion introduced by the refraction of light rays, we compute the intersection of our 3D rays to a sphere of radius *R* from the center of the camera system (Figure 19). From the equations of a ray in space and a sphere [77] we have
(21)$$\left(D\cdot D\right)L^{2}+2D\cdot \left(O-C\right)L+\left(O-C\right)\cdot \left(O-C\right)-R^{2}=0$$
where *D* is the ray’s direction, *O* is the origin of the ray, *C* is the center of the sphere, *R* corresponds to the radius of the sphere, and *L* represents the length of the ray. If we solve this equation for *L*, we can compute a 3D point on the sphere using the equation of a ray:(22)p=O+L∗D

Equation (Equation 22) computes the 3D location of each pixel as if they were located on the surface of a sphere with radius *R*. Then, since we know the camera calibration parameters, the 3D points can be projected to a 2D image plane by
(23)$$x=PX_{sphere}$$
where $P=K\left[R|T\right]$ is composed of the intrinsic and extrinsic parameters.

As described above, the projection of 3D points will produce a distorted image of the scene, shown in the left representation of Figure 20. To correct the image, we locate the four mid-points of the current distorted image and register them into a rectangle of new image size dimensions (e.g., half the original image width and height). To find a homography that registers the distorted image into an undistorted image, we use a least squares procedure:(24)$$Hu=b\Rightarrow \left[\begin{array}{ccc} a & 0 & c\\ 0 & a & d\\ 0 & 0 & 1 \end{array}\right]\left[\begin{array}{c} u\\ v\\ 1 \end{array}\right]=\left[\begin{array}{c} u^{\prime }\\ v^{\prime }\\ 1 \end{array}\right]$$

Then, the new intrinsic matrix of the camera is defined by applying the homography to the original intrinsic parameters:(25)$$K_{u}=H*K$$

Finally, these camera parameters are used to compute the new 2D locations of the 3D points by using Equation (Equation 23). The process described above is required to remove distortion from the omnidirectional images from all cameras. It should be noted that this process can be computationally expensive and add latency to the image acquisition. Therefore, we used the 2D points computed from the projection process to create look-up tables that map the original location of a pixel into the corresponding location in the undistorted image. In this way, the computational time to process the image is improved, and the acquisition time falls in the range of 30 to 70 ms.

### 5.5. Evaluated Setups

The deployment of the system was tested with two different configurations of the testing pool, to evaluate the response of our robot in specific scenarios. In the first configuration, some texture was added to the walls of the pool and the AUV was piloted towards them. The trials were performed at different speeds to observe the effect of speed on the calculated risk of the 3D point cloud and its impact on the resultant force. A second setting consisted of a set of obstacles placed in the pool to test how the system reacts when navigating through narrow scenarios (an illustrative video of online tests can be found as Appendix A to this work, or at https://www.youtube.com/watch?v=kI_uTeURSq0 (accessed on 12 June 2022)). Both results are described below.

#### 5.5.1. Velocity Changes

The response of the system to changes in velocity was tested to verify the performance of the risk computation and the effects of the resulting output force. The tests were performed by moving the robot towards the textured walls of the controlled environment from the same starting point and with different speed settings. The Girona 1000 was set to maximum speeds of 0.5×, 1.5×, and 2× its nominal speed. Similarly to the offline tests, we observed that the risk for points in the heading direction is higher when the robot’s velocity increases, so obstacles are perceived as more dangerous. On the other hand, the system can detect this change and sends a warning signal (resultant repulsive force) to the robot’s controller so that it can take evasive action at an early stage. This situation is illustrated in Figure 21, where two speed setups are shown. The figure shows the time when the robot was stopped by the controller due to the warning signal sent to the AUV. It can be seen that as the speed increases, the Girona is stopped at an earlier distance, 0.5 m before, to the place where it was stopped at nominal speed.

#### 5.5.2. Navigation through Narrow Passages

To assess the reliability of the system in maintaining the safety of the robot, the proposed framework was also tested in a third complex scenario where objects were placed in the environment, and the AUV was controlled to navigate through narrow passages. To emulate real-time conditions, the Girona 1000 was freed from speed constraints, meaning that the operator could regulate the robot’s speed via commands from a controller. Several trials were performed to determine the responses of the system. Figure 22 shows a frame sequence in which the robot was navigated through these structures. The images illustrate how at a starting point when the Girona is not yet inside the passage (Figure 22a), most of the points have low-risk values (depicted as blue color zones). However, when the robot enters the narrow space (Figure 22b–d), we can observe how points on the surface of the cylinder have medium risk values (cyan and yellow zones), and the repulsive force also starts increasing, telling the robot to be aware along its sides. To visualize the position relationship of the Girona 1000 with respect to the obstacles, the objects have been overlaid in the point cloud images which are displayed in the second row of Figure 22. In this trial, the operator is still able to control the heading of the robot as the force has not reached its threshold value, which would result in the system taking control to perform evasive movements. Therefore, the tests show how the system lets the AUV move in a complex scenario while maintaining spatial awareness of the local surroundings.

## 6. Conclusions

We presented an omnidirectional multi-camera system for early collision detection and avoidance for underwater vehicles, capable of outputting warning signals that can be easily used by an operator or a control system to perform evasive maneuvers. Our system uses the MKFs and pose estimates of an SLAM system (Multi-Col SLAM) to create a denser 360° map representation of the local environment and assess the risk of obstacles based on this representation. This assessment is later used to compute a resulting repulsive force that is used to move the robot away from potential collisions, and an equirectangular risk map representation. Since the risk is evaluated based on the trajectory of the robot, the calculation of the repulsion force naturally takes into account conditions such as the water flow or currents so that the control systems can react to hazards and ensure the safety of the robot. On the other hand, the 2D omnidirectional risk map allows pilots to easily understand and perceive the zones that could lead to potential collisions. Moreover, we implemented such a framework in an AUV (Girona 1000) and performed real-time tests. The results show that obstacles near the robot and along the current trajectory were assigned higher risk values. Furthermore, these values are proportionally dependent on the speed of the system and the cautioning risk cone. We show that the system can correctly detect when surrounding obstacles become a threat to the robot and react to them. Dangerous zones appear when the robot approaches an obstacle that is on its motion path. Finally, we successfully demonstrated that the system can be used to navigate an AUV through narrow passages by avoiding collisions. To our knowledge, no system with the presented capabilities has yet been implemented and tested for ROV/AUV applications, and this work constitutes a first achievement in this direction.

## Figures and Tables

**Figure 1 sensors-22-05354-f001:**
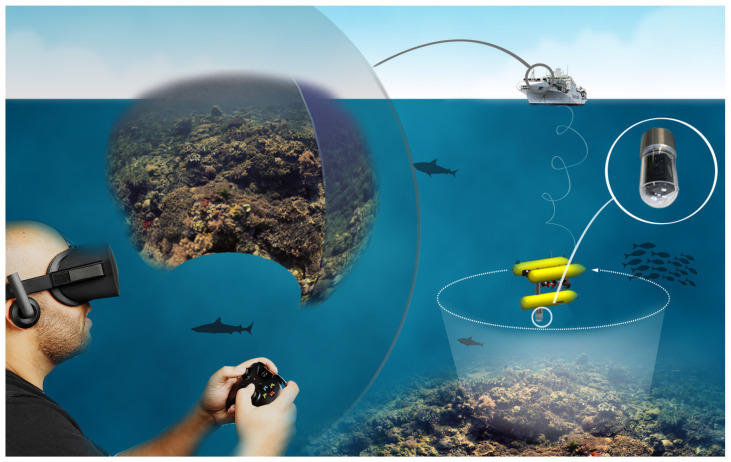
Envisioned ROV piloting system, where early collision detection work of this paper is a central part.

**Figure 2 sensors-22-05354-f002:**
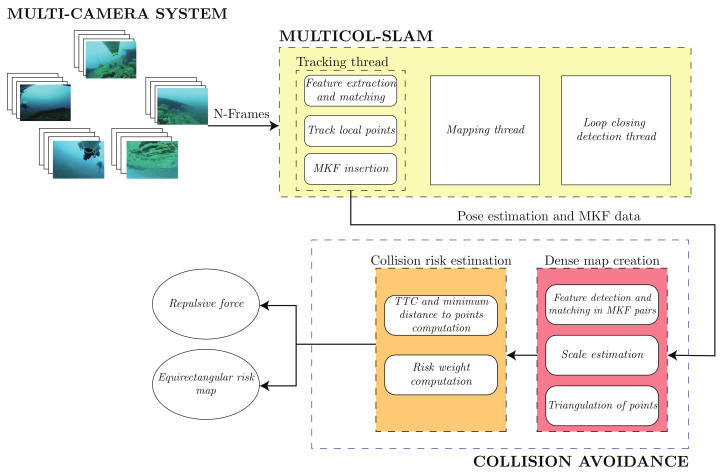
Collision detection thread work flow.

**Figure 3 sensors-22-05354-f003:**
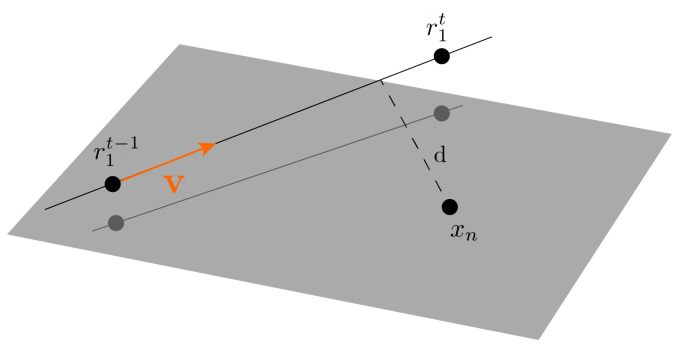
Distance *d* from a point $x_{n}$ in 3D space to a line.

**Figure 4 sensors-22-05354-f004:**
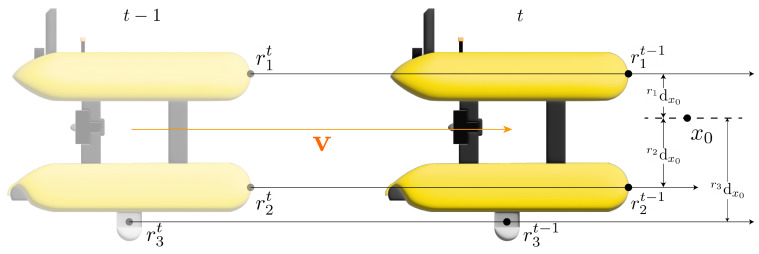
Computation of the distance ${}^{r_{i}}d_{x_{n}}$ measurements for a specific point $x_{0}$. The scheme shows a movement of the robot from a previous time t−1 (transparent frame) to a current time *t* (opaque frame), and the parameters *d* and *t* are now calculated for multiple selected points on the robot’s body ($r_{1}$,$r_{2}$,$r_{3}$).

**Figure 5 sensors-22-05354-f005:**
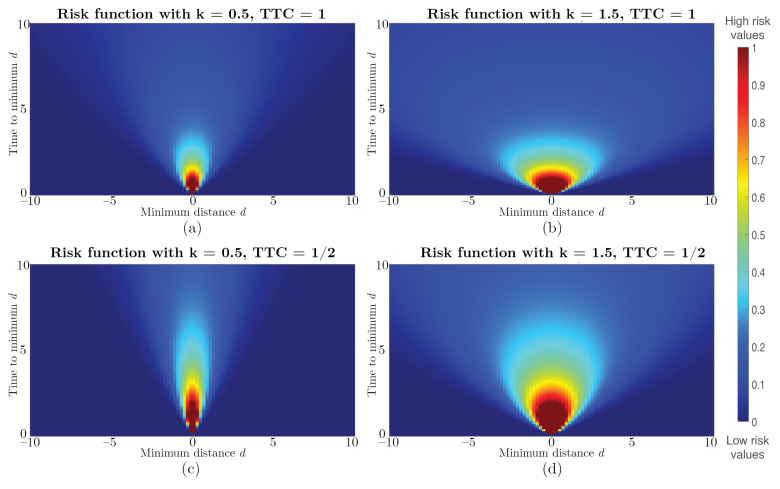
Graphical visualization of the effects of the risk function parameters. Panels (**a**,**b**) show the effects of changing the *k* parameter in the risk function, resulting in a wider window of caution as *k* increases. The lower images (**c**,**d**) show the same configuration, but with the effects of changing the TTC parameter (increasing speed), resulting in (temporally) more distant objects being perceived as dangerous than in the corresponding images (**a**) or (**b**).

**Figure 6 sensors-22-05354-f006:**
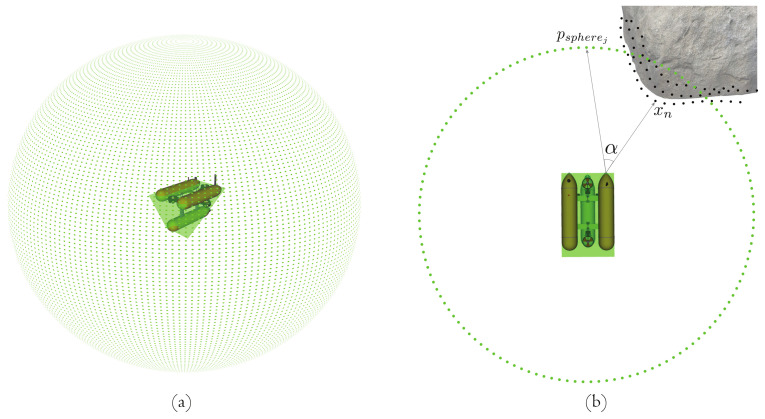
Force computation for each obstacle 3D point. (**a**) The ROV is surrounded by a sphere of points (green points) in order to have a constant number of areas in each computation. (**b**) Every detected point in the local map (black points) is then associated with a region point in the sphere by computing their dot product and taking the vector that is closest to the direction of $\vec{f_{x_{n}}}$.

**Figure 7 sensors-22-05354-f007:**
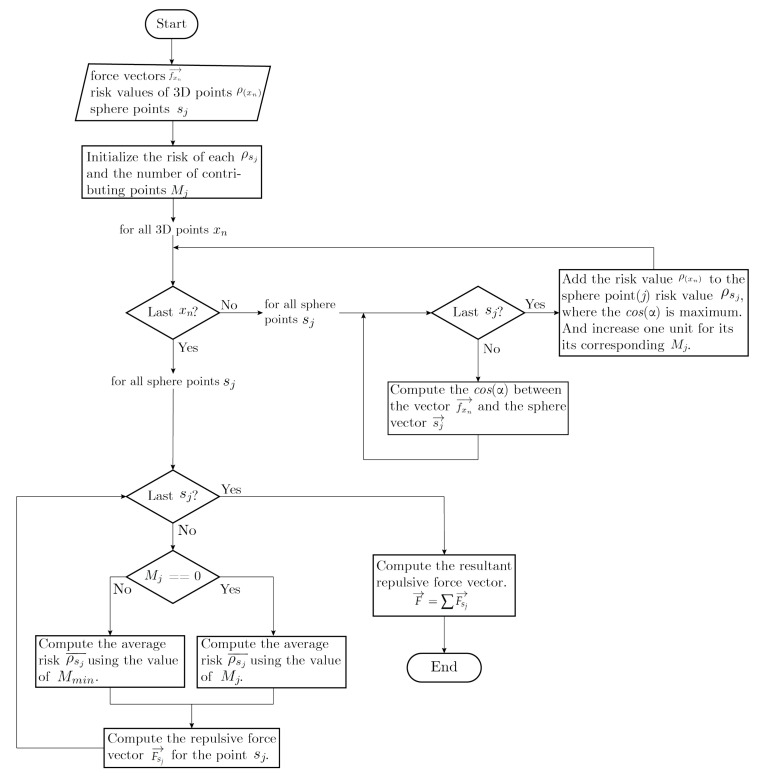
Flowchart of the process described in Algorithm 1.

**Figure 8 sensors-22-05354-f008:**
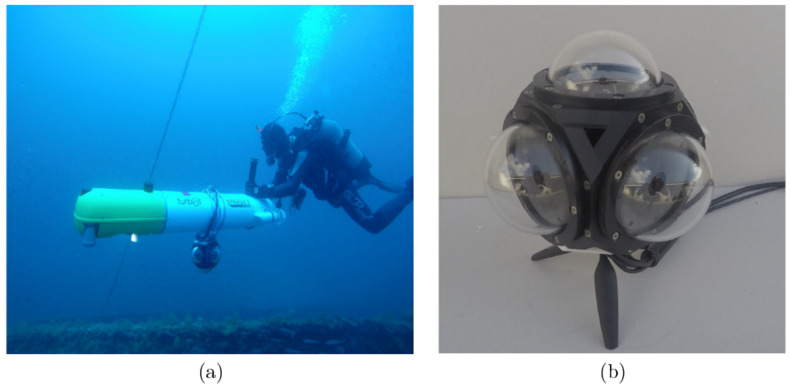
Setup for collecting the Boreas dataset. (**a**) MCS mounted on Sparus II AUV. (**b**) Omnidirectional camera system composed of five GoPro Hero 4 cameras.

**Figure 9 sensors-22-05354-f009:**
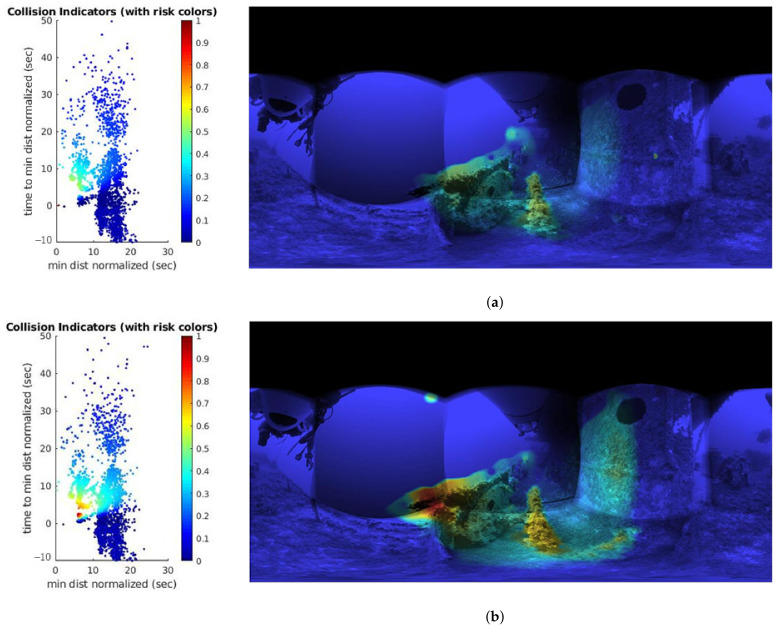
Effects on the estimated risk when varying the parameter *k*. The plots on the left illustrate the distributions of the detected 3D points, represented in a plot of $d_{norm}$ versus ${\mathrm{TTC}}_{i}$ and color-coded by the associated risk value, while the images on the right are equirectangular views with color-coded risk. (**a**) Parameter *k* = 1.2. (**b**) Parameter *k* = 2.

**Figure 10 sensors-22-05354-f010:**
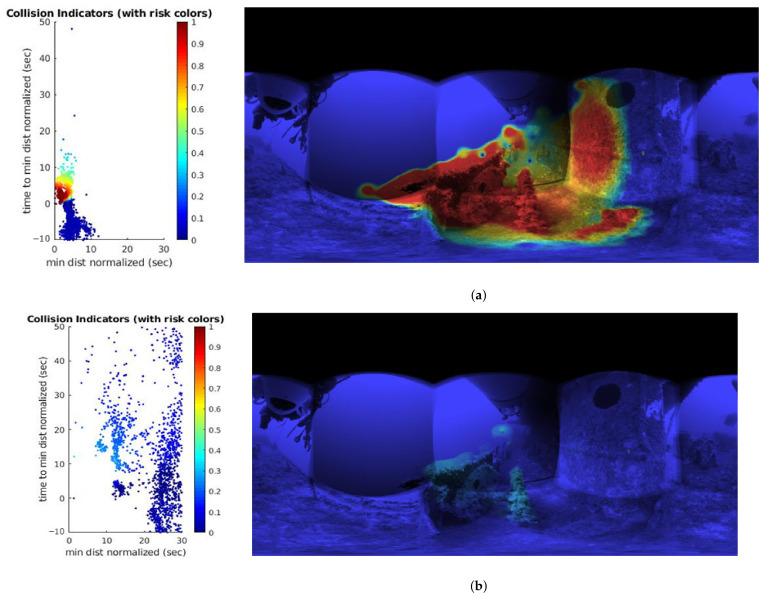
Effects on the risk when the velocity changes and k=1.2. (**a**) Velocity increased by a factor of four. (**b**) Reducing the velocity by half.

**Figure 11 sensors-22-05354-f011:**
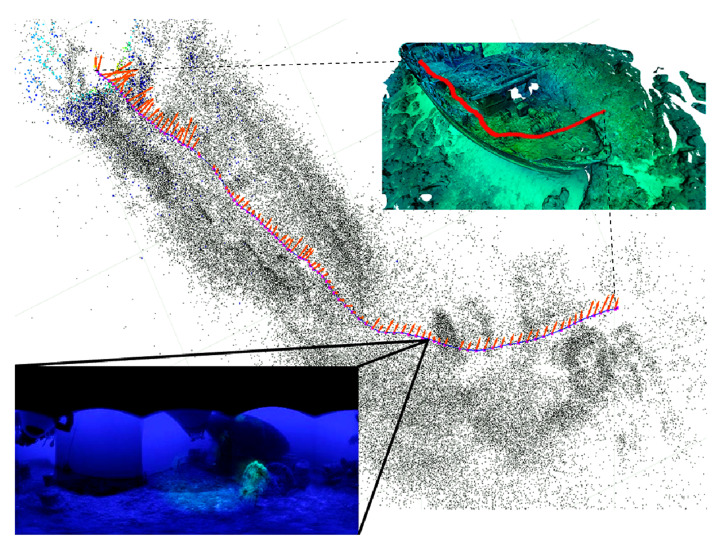
Vehicle trajectory and repulsion forces (orange arrows) while navigating close to the Boreas shipwreck. The top inset is a 3D reconstruction [67] viewed from the same approximate angle as the point cloud. The black points represent previous computed point clouds through the trajectory, and the last position shows the corresponding point cloud with the risk associated to each 3D point (colored points). The bottom inset contains the omnidirectional camera view with the color-coded risk overlay.

**Figure 12 sensors-22-05354-f012:**
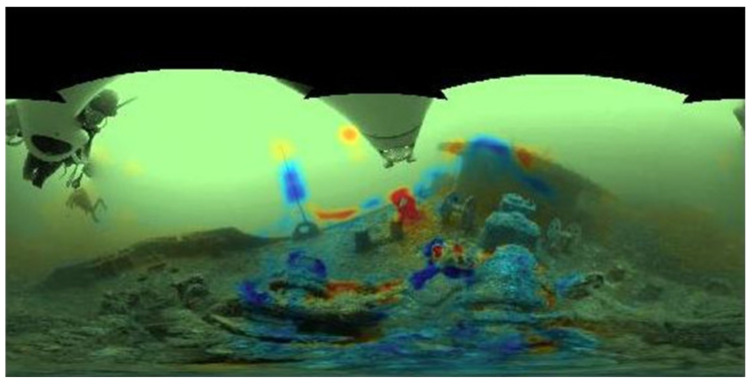
Color -coded difference between the calculated system risk map and the one from the ground truth. Green values correspond to no differences, red values to false positive detections, and blue values to false negatives. The other colored areas are regions where the difference between our algorithm and the ground truth is close to each other and they are not considered for the analysis.

**Figure 13 sensors-22-05354-f013:**
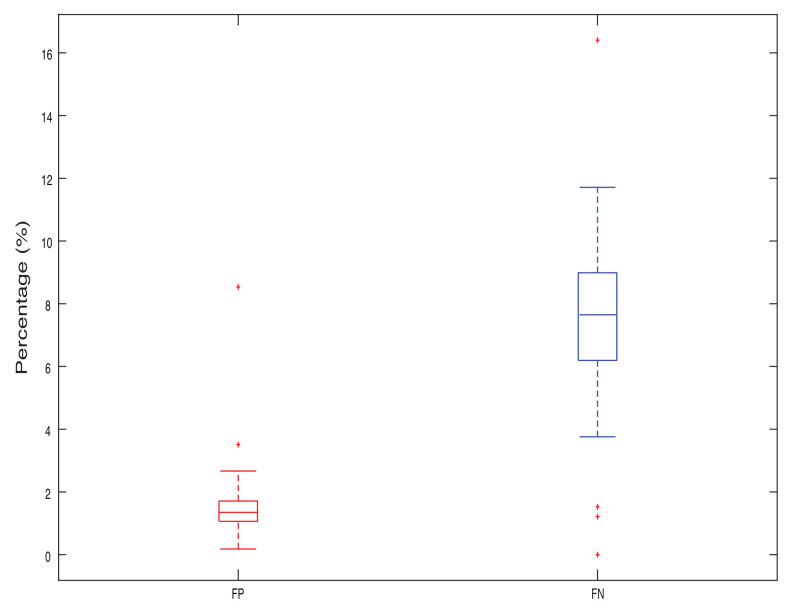
Performance in terms of percentage of false positives and false negatives for the Boreas dataset. The red + symbols represent the outliers found in both measures.

**Figure 14 sensors-22-05354-f014:**
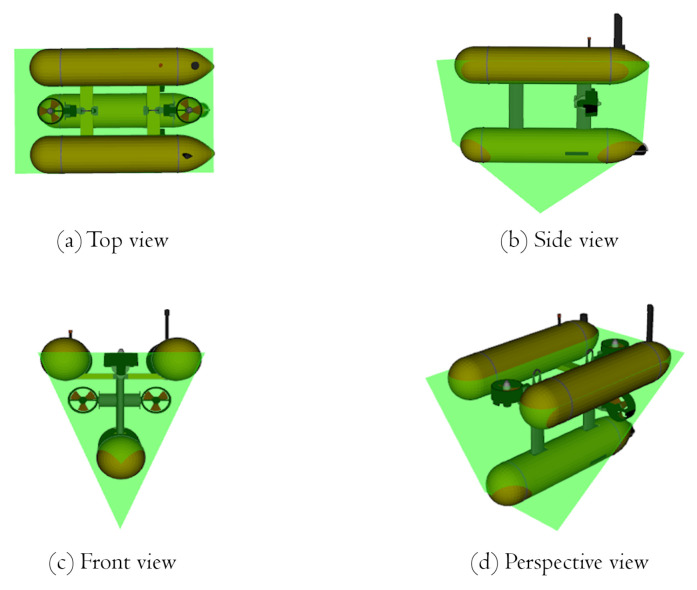
Views of the safety cage (green), which was created using six points on the robot chassis and the multi-camera housing (seven points in total) to calculate the collision risk.

**Figure 15 sensors-22-05354-f015:**
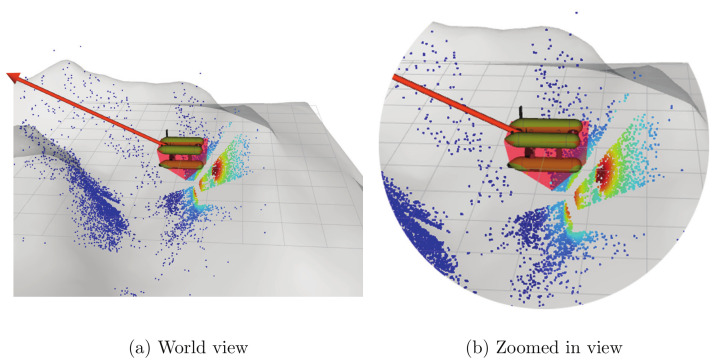
Case in which the risk calculation identifies several danger zones (red zones) for the robot. The picture on the left shows the world view of the system at that particular time, while the right image is a zoomed-in version of the points closest to the robot. The zones where the risk is low are represented by blue areas in the images, and areas that turn gradually into a reddish color represent higher risk areas. Two danger zones (shown in red) are detected while the robot is moving towards the wall. The direction and magnitude of the repulsive force are represented by the red arrow.

**Figure 16 sensors-22-05354-f016:**
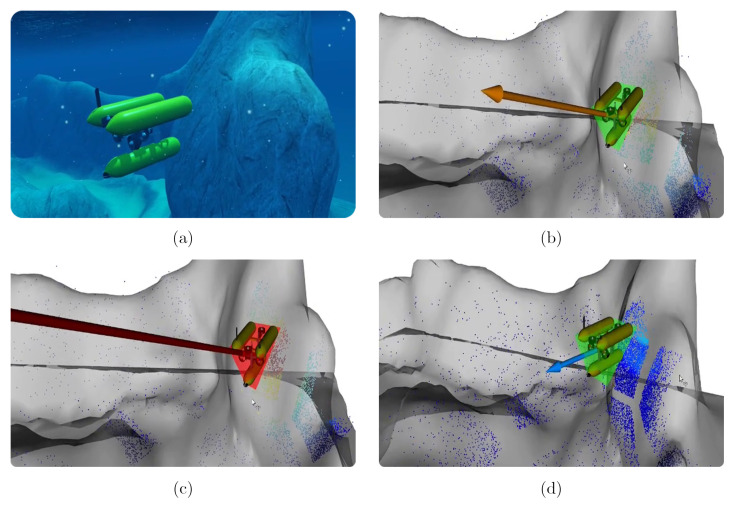
Representation of the behavior of the system when the robot is near a danger zone. (**a**) A capture of the exploration performed in the Stonefish simulator. Images (**b**–**d**) are a series of consecutive frames that show how the system reacts to a possible danger. The zones where the risk is low are represented by blue areas and areas that turn gradually into a reddish color represent higher risk areas. Figure (**c**) shows the moment when the risk of collision with an obstacle is so large that a control system takes control of the robot and moves it away from that zone.

**Figure 17 sensors-22-05354-f017:**
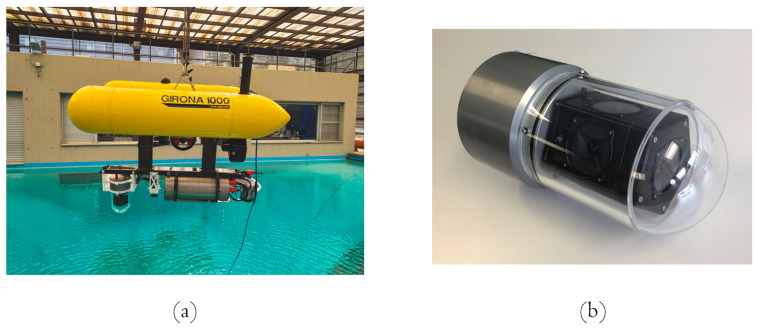
System configuration for real-time deployment. (**a**) Girona 1000 AUV coupled with the camera for testing. (**b**) Ladybug 3 multi-camera system inside our manufactured waterproof housing.

**Figure 18 sensors-22-05354-f018:**
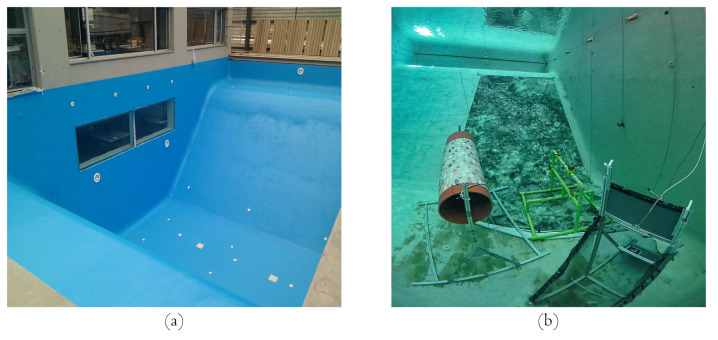
Testing pool setup. (**a**) Visualization of the testing pool characteristics. (**b**) Configuration of the testing environment, which is provided with texture on the walls and objects located around the bottom.

**Figure 19 sensors-22-05354-f019:**
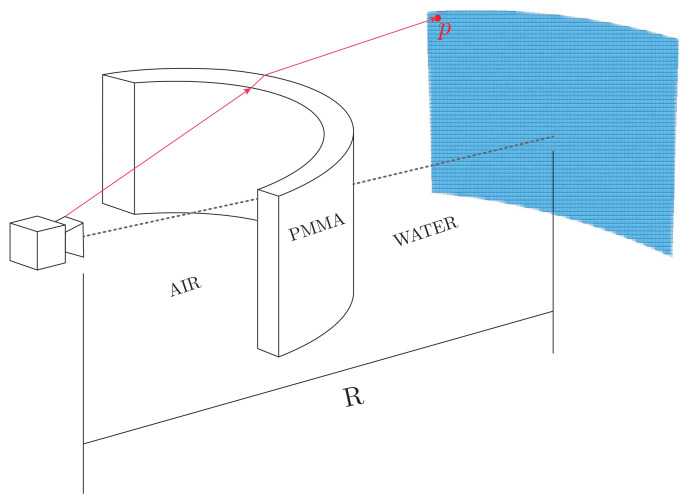
Ray tracing schematic of a single ray being traced to a sphere of radius *R* to produce the underwater image. The ray direction is changed twice when it passes through different media.

**Figure 20 sensors-22-05354-f020:**
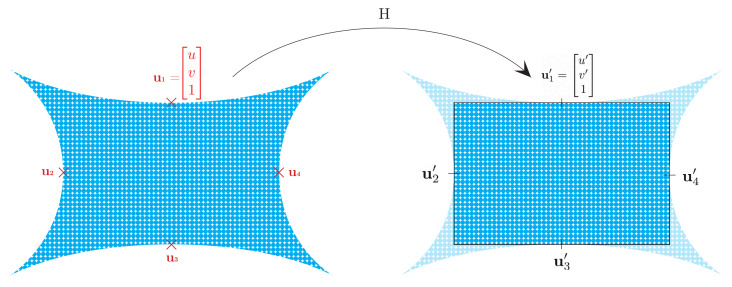
Image correction process. The distorted image (left) generated by projection of 3D points is registered into a rectangular area containing a new undistorted image; this registration is obtained by computing the homography matrix *H*.

**Figure 21 sensors-22-05354-f021:**
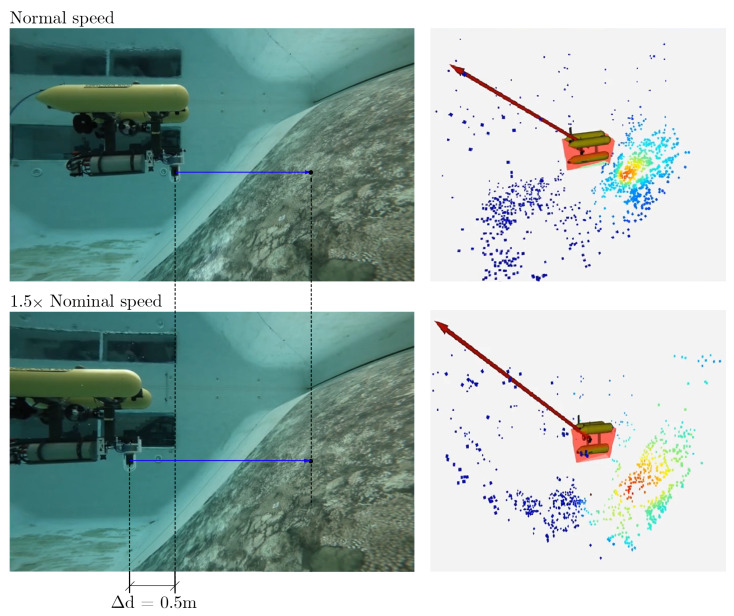
Performance of the robot as velocity changes. In the upper part of the figure, the behavior of the system at the nominal speed of the Girona is depicted. The bottom pair of images show how the system detects a larger dangerous area (red zone) when the speed is increased by a factor of 1.5 units. This change makes the AUV stop a Δd distance before compared to the nominal speed condition.

**Figure 22 sensors-22-05354-f022:**
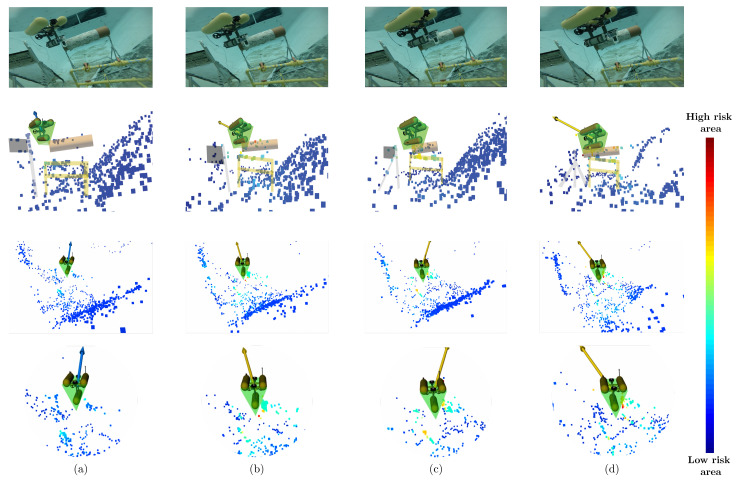
Test scenario where the Girona 1000 AUV was controlled to pass through a narrow passage created by the set obstacles. Images from (**a**–**d**) show a sequence of frames with their respective point cloud representation, and the risk calculated for each obstacle. Frames (**c**,**d**) depict medium-danger zones (yellow areas) where the AUV has become too close to the pipe, and the force is directed outwards from that area.

## Data Availability

No new data were created or analyzed in this study. Data sharing is not applicable to this article.

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
