# Peer review of "Collision Detection and Avoidance for Underwater Vehicles Using Omnidirectional Vision"

_sensors, 2022, doi:10.3390/s22145354_

Round 1

Reviewer 1 Report

This paper presented an omnidirectional multi-camera early collision detection and avoidance system, a novel approach of underwater vehicle with capacities for navigation in complex environments was provided, a visual SLAM framework by creating a 360° real-time point was created, and a strategy to access the risk of obstacles was developed. It is an excellent work for constructing two phases of experiments to evaluate the system capabilities. To improve the quality of this manuscript in essence, the following issues need be explained or responded:

1. It is suggested to change the expression form of algorithm 1 into a flow chart, which could be more concise and intuitive.

2. About the equation 20, it is necessary to specify what the symbols denote.

3. The symbol t represents the time in the previous section, while it denotes the length of the ray in the process of correcting the distortion introduced by the refraction, which is very ambiguous.

4. The pictures of the Girona 1000 passing through the narrow passage shown in figure 20, where the position relationship of the AUV is not obvious.

5. When the AUV passes through the narrow passage, the performance of the system in response to speed changes will be more important, it is suggested to supplement the test of passing experiments at different speeds.

6. Whether the experiments considers the influence of water flow, and the flow condition is critical to the practicality of the system framework.

7. The abstract notes that the paper presented an omnidirectional multi-camera early collision detection and avoidance system for ROVs, however the test verification is carried on an AUV, whether the difference in vehicle type will make a difference?

Reviewer 2 Report

This manuscript was shown good results for Collision Detection and Avoidance for Underwater Vehicles Using Omnidirectional Vision. 

I have several comments as,

1. Please cite each conclusion and equation if not yours

2. In the video results, Please show how to synchronize all camera data, and what is the delay of all the cameras compared to a ref camera?

3. Show your time complexity in a table for the pose estimation and mapping?. What is the period of the camera? ( ladybug 3 is quite slow in Ubuntu with ROS driver, can you solve the problem?)

4. What different V-SLAM methods are compared to ORB-SLAM2, openVslam, omniSLAM, etc...

Reviewer 3 Report

It is a well-prepared and presented manuscript. There is no additional comment from my side. I did enjoy the reading. 

Reviewer 4 Report

This paper presents an interesting results on collision detection and avoidance for Underwater Vehicles. The field data set results were solid  But content about  the Collision avoidance approach need more sound proof and analysis.

1.  in line  273-275, ,local mapping thread extends the map by creating new points and deleting all redundant  map points and MKFs. The mapping thread also maintains the consistency of the global  map by performing a global Bundle Adjustment optimization step. 

   How to get MKFs for video sequence, and that is an important step need some context to complete that.

2. For the section 4.3 Risk estimation,  It will be better to present some quantitative examples of parameters. 

Round 2

Reviewer 2 Report

I have no further comments, but I still wonder if the comparison results were not presented.